# The Role of Necroptosis in ROS-Mediated Cancer Therapies and Its Promising Applications

**DOI:** 10.3390/cancers12082185

**Published:** 2020-08-05

**Authors:** Sheng-Kai Hsu, Wen-Tsan Chang, I-Ling Lin, Yih-Fung Chen, Nitin Balkrushna Padalwar, Kai-Chun Cheng, Yen-Ni Teng, Chi-Huei Wang, Chien-Chih Chiu

**Affiliations:** 1Department of Biotechnology, Kaohsiung Medical University, Kaohsiung 807, Taiwan; b043100050@gmail.com; 2Department of Medical Laboratory Science and Biotechnology, Kaohsiung Medical University, Kaohsiung 807, Taiwan; linili@kmu.edu.tw; 3Department of Surgery, School of Medicine, College of Medicine, Kaohsiung Medical University, Kaohsiung 807, Taiwan; wtchang@kmu.edu.tw; 4Division of General and Digestive Surgery, Department of Surgery, Kaohsiung Medical University Hospital, Kaohsiung 807, Taiwan; 5Center for Cancer Research, Kaohsiung Medical University, Kaohsiung 807, Taiwan; 6Department of Laboratory Medicine, Kaohsiung Medical University Hospital, Kaohsiung 807, Taiwan; 7Graduate Institute of Natural Products, Kaohsiung Medical University, Kaohsiung 807, Taiwan; yihfungchen@kmu.edu.tw; 8Department of Chemistry, National Institute of Technology Tiruchirappalli, Tiruchirappalli 620015, Tamilnadu, India; nitinbpadalwar@gmail.com; 9Department of Ophthalmology, Kaohsiung Municipal Hsiaokang Hospital, Kaohsiung 812, Taiwan; pington64@gmail.com; 10Department of Ophthalmology, Kaohsiung Medical University Hospital, Kaohsiung 807, Taiwan; 11Department of Biological Sciences and Technology, National University of Tainan, Tainan 700, Taiwan; tengyenni1968@gmail.com; 12Department of Biological Sciences, National Sun Yat-sen University, Kaohsiung 804, Taiwan; 13Department of Medical Research, Kaohsiung Medical University Hospital, Kaohsiung 807, Taiwan; 14The Graduate Institute of Medicine, Kaohsiung Medical University, Kaohsiung 807, Taiwan

**Keywords:** necroptosis, reactive oxygen species, cancer, chemotherapy

## Abstract

Over the past decades, promising therapies targeting different signaling pathways have emerged. Among these pathways, apoptosis has been well investigated and targeted to design diverse chemotherapies. However, some patients are chemoresistant to these therapies due to compromised apoptotic cell death. Hence, exploring alternative treatments aimed at different mechanisms of cell death seems to be a potential strategy for bypassing impaired apoptotic cell death. Emerging evidence has shown that necroptosis, a caspase-independent form of cell death with features between apoptosis and necrosis, can overcome the predicament of drug resistance. Furthermore, previous studies have also indicated that there is a close correlation between necroptosis and reactive oxygen species (ROS); both necroptosis and ROS play significant roles both under human physiological conditions such as the regulation of inflammation and in cancer biology. Several small molecules used in experiments and clinical practice eliminate cancer cells via the modulation of ROS and necroptosis. The molecular mechanisms of these promising therapies are discussed in detail in this review.

## 1. Introduction

In recent decades, the biological characteristics of apoptosis have been well established; this has resulted in an increase in apoptosis-associated drugs [1,2]. For instance, previous evidence has indicated that diosgenin, a molecule extracted from a traditional Chinese herb (Trigonella foenum-graecum), has proapoptotic effects against different types of cancer, such as Her-2-positive breast cancer, hepatocellular carcinoma and skin squamous cell carcinoma [3,4,5]. Nevertheless, emerging studies have reported that several cancer cell lines such as pancreatic cancer cells and melanoma cells are drug-resistant. These cells become less apoptotic and less sensitive to chemotherapies and develop mutations in pro-apoptotic genes [6,7,8]. Preliminary evidence has shown that breast cancer cell lines, such as MCF-7 and MDA-MB-231, are resistant to chemotherapy (CT) drugs (e.g., docetaxel) due to the upregulation of miRNA-34a and miR-141, which impedes the apoptotic signaling pathway [9,10]. Another study suggested that the efficacy of several primary chemotherapies (e.g., cisplatin, gemcitabine, paclitaxel and 5-fluorouracil) is limited owing to the upregulation of anti-apoptotic proteins, such as Bcl-2 and Bcl-xL, and mutations in TP53, Fas or Bax [2,11,12]. Furthermore, Bonapace et al. confirmed that some patients with intractable acute lymphoblastic leukemia show higher resistance to glucocorticoids—first-line CT drugs—due to increased levels of the anti-apoptotic protein myeloid cell leukemia sequence 1 (MCL-1) [13].

Bypassing apoptosis-induced cell death is a mechanism of multidrug resistance; hence, the modulation of necroptosis, a caspase-independent mechanism of programmed cell death, seems to be a promising approach to overcoming apoptotic drug resistance [14,15,16]. In addition to being caspase-independent, necroptosis possesses different features from apoptosis. For instance, necroptotic cell death can trigger an inflammatory response [17]. It can also contribute to lysosomal and plasma membrane permeabilization without causing chromatin condensation, cytoplasmic shrinkage or nuclear fragmentation [18,19].

Because the development and approval of new drugs requires a considerable amount of time and carries a great cost, drug repurposing has gained popularity in the field of cancer treatment. Several anti-cancer drugs have been identified: (1) Ponatinib, a BCR-ABL tyrosine kinase inhibitor that is used for patients with chronic myeloid leukemia (CML), has great potential to block necroptosis [20,21]. (2) Pazopanib, a receptor tyrosine inhibitor with multiple targets that has been approved and used for patients with advanced renal cell carcinoma, has also been shown to block necroptosis [21]. In conclusion, the development of novel therapies targeting necroptosis is beneficial.

## 2. The Role of Reactive Oxidative Species (ROS) in Human Physiology and Cancer Biology

ROS are specific molecules that contain oxygen (e.g., free radicals) converted from molecular oxygen, and they are mainly generated within mitochondria [22]. Electrons flow through the mitochondrial electron transport chain (ETC) to reduce molecular oxygen to superoxide (O2•).

ROS, especially O2•, are metabolites of oxidative respiration; oxidative phosphorylation (OXPHOS) takes place within four transmembrane proteins embedded in the inner mitochondrial membrane, including complexes I, II, III and IV. O2• produced by complexes I and II is primarily located in the mitochondrial matrix, while complex III can generate O2• in both the intermembrane space and matrix. Subsequently, O2• within the matrix is reduced to hydrogen peroxide (H_2_O_2_) by superoxide dismutase protein 2 (SOD2) [23]; however, O2• within the intermembrane space needs to be translocated to the cytosol and converted into H_2_O_2_ by superoxide dismutase protein 1 (SOD1). This process initiates cell signaling events [24,25]. However, ROS has dual effects. On the one hand, ROS can trigger biological processes; for instance, at low concentrations, they induce cell proliferation. On the other hand, at high levels, they may damage DNA, protein and lipids and cause cell death [26]. H_2_O_2_ is reduced to H_2_O via peroxidases to protect cells from these deleterious effects [27]. Under physiological conditions, ROS eliminate harmful microorganisms (e.g., infectious bacteria) and subsequently initiate cell signals to repair damaged epithelium [27,28]. Moreover, when ROS levels are low, they can induce stress responses such as ER stress to promote cell survival and modulate energy metabolism, apoptosis and the inflammatory response [29]. ROS derived from the complex III of the ETC are necessary for CD4^+^ T lymphocyte activation and antigen-specific CD4^+^ and CD8^+^ T lymphocyte expansion [30].

Unfortunately, the aberrant modulation of ROS generation can drive tumorigenesis and facilitate cancer progression. ROS produced by mitochondrial complex III have been reported to stabilize hypoxia-inducible factor-1alpha (HIFα) under hypoxic conditions [31]. The stabilization of HIFα is key for the upregulation of vascular endothelial growth factor (VEGF), which can promote endothelial cell proliferation and angiogenesis [32,33]. Recent evidence has suggested that associations among ROS, the β-catenin-dependent Wnt pathway and c-Myc promote tumor proliferation [34]. In addition, ROS also activate the extracellular-signal-regulated kinase/mitogen-activated protein kinase (ERK/MAPK) pathway to induce cell proliferation and survival [27]. Besides, Weinberg et al. reported that mitochondrial ROS could modulate K-Ras-mediated anchorage-independent growth via the MAPK/ERK1/2 pathway [35] (Figure 1).

## 3. Molecular Mechanisms Underlying Necroptosis and Cell Death Signaling Pathways

Necroptosis is initiated by the engagement of death receptors and their corresponding ligands, genotoxic stresses or anti-tumor drugs (e.g., etoposide) [36,37]. Three membrane-embedded death receptors—TNFR1, Fas, and death receptor 4 and 5 (DR4/5), all of which belong to the tumor necrosis factor receptor (TNFR) superfamily—play key roles in cell death signaling pathways and are discussed in detail below. First, a signaling pathway that shares similar components with apoptosis and cell survival is triggered by the interaction between TNFR superfamily members and TNF-α—the primary mediator of cell death, including both apoptosis and necroptosis [2,38,39].

Among death receptors, TNF-α is the most investigated for its role in mediating necroptosis [2], and TNFR1 serves as a trigger of inflammation, apoptosis and necroptosis [40]. In the TNF-α-induced signaling pathway, the binding of TNFR1 with its ligand—TNFα— leads to the assembly of complex I, which contains TRADD, TRAF2, CYLD, cIAP1 and RIP1. RIP1 is polyubiquitinated by cIAP1 at the lysine (K) 63 residue, resulting in the recruitment of the IKK complex (composed of two kinases, IKKα and IKKβ, and a regulatory subunit, NEMO/IKKγ) and TGF-β-activated kinase 1 (TAK1), which respectively induce the NF-κB and MAPK signaling pathways to promote cell survival and inflammation [36]. Furthermore, the process mentioned above results in the blockade complex IIa, which consists of caspase-8, FADD, RIP1, and complex IIb, which involves caspase-8, FADD, RIP1, RIP3 and mixed lineage kinase domain-like (MLKL); the blockade of these complexes contributes to the inhibition of RIP1-mediated apoptosis or necroptosis [41]. By contrast, when RIP1 is deubiquitinated by CYLD (a K63-specific deubiquitinating enzyme) rather than polyubiquitinated, the formation of complexes IIa and IIb is promoted. This process has two ramifications—apoptosis and necroptosis. When caspase-8 is intact, it cleaves RIP3, a key kinase that promotes necroptosis, which blocks the packaging of complex IIb, also known as the necrosome, and favors apoptosis. By contrast, RIP3 can be phosphorylated and protected from degradation by caspase-8 inhibitors (e.g., z-Val-Ala-Asp fluromethyl ketone (z-VAD-fmk)) or inactive caspase-8, inducing necroptosis and related downstream events such as ROS burst [42], cytosolic ATP reduction and plasma membrane permeabilization [2,43]. Subsequently, RIP3 phosphorylation can induce the phosphorylation of downstream MLKL; activated MLKL translocates to the plasma membrane, enhancing the permeability of the membrane to execute cell death [44,45].

Fas, also called CD95, is another significant death receptor that plays a key role in the activation of apoptosis and immune surveillance, such as the elimination of aberrant cells, virus-infected cells and self-reactive lymphocytes [46]. It is activated via its interaction with its ligand, FasL, and subsequently recruits FADD, RIP1, procaspase-8 and cellular FADD-like IL-1β-converting enzyme-inhibitory protein (cFLIP) to form death-inducing signaling complex (DISC). In addition, cFLIP plays a pivotal role in determining cell fate [37]. On the one hand, heterodimers consisting of FADD and cFLIP inhibit DISC and induce NF-κB, promoting cell survival and cytokine production instead of apoptosis [47]. On the other hand, pro-caspase-8 homodimers stabilize DISC and activate caspase-8 [48]. Moreover, when caspase-8 is intact, the Fas signaling pathway can also lead to RIP1-dependent cell death through the formation of complex IIa.

DR4/5, the other primary death receptor of the cell death signaling pathway, consists of two death receptors, tumor necrosis factor-related apoptosis-inducing ligand (TRAIL) receptor 1 (TRAILR1) and TRAIL receptor 2 (TRAILR2), and interacts with its ligand TRAIL (also known as Apo2L); this may promote the formation of complex IIa or complex IIb. Intact caspase-8, which can inhibit RIP3 and FADD, favors apoptosis; however, caspase-8 inactivation leads to the formation of complex IIb, triggering downstream signaling processes, such as RIP3 and MLKL phosphorylation, and facilitating necroptosis [2,49] (Figure 2).

## 4. The Role of Necroptosis in Human Physiology and Cancer Biology

### 4.1. The Role of Necroptosis in the Modulation of Immunity

Necroptosis, a form of programmed cell death, is triggered by death receptor signaling and participates in several physiological conditions; one salient example is the defense mechanism against intracellular infection [50]. Necroptosis also refers to a particular type of regulated necrosis that depends on the serine/threonine kinase activity of RIP1 [51]. In comparison with apoptosis—which is characterized by non-inflammatory cell death, including chromatin condensation, the contraction of cellular volume and the release of intact cellular contents into membrane-wrapped vesicles (apoptotic bodies)—necroptosis features the initiation of immunity, which is conducive to the clearance of aberrant cells [52].

In addition to eliminating anti-apoptotic cancer cells, necroptosis also plays a pivotal role in adaptive immunity; it modulates the antigen-induced proliferation of T lymphocytes, which is important for T lymphocyte homeostasis and survival [53]. Through primary induction via TNFR1, several anti-cancer drugs promote autocrine TNF production through necroptosis, enhancing immune function [54]. For instance, neoalbaconol, a natural compound extracted from the mushroom *Albatrellus confluens*, induces necroptosis through autocrine TNF production [55]. As mentioned earlier, necroptosis can be triggered when caspase-8 is defective. A previous experiment reported that in mice that lack caspase-8 or are unable to recruit caspase-8 to T lymphocytes, the execution of normal immune functions fails. In other words, defective caspase-8 leads to dysfunctional T lymphocyte homeostasis and a decreased number of peripheral T lymphocytes [56]. However, another study showed that the defective accumulation of activated T lymphocytes resulting from the absence of caspase-8 and the induction of necroptosis can be rescued via the necrostatin-1 (Nec-1)-induced inhibition of necroptosis [53,57].

### 4.2. The Immunosurveillance of Necroptosis in Cancer

Necroptosis plays a pivotal role in the modulation of human immunity. For instance, a previous experiment conducted by Aaes et al. indicated that necroptotic cells can release DAMPs such as HMGB1 and ATP and chemokines (e.g., CXCL1). This stimulated bone-marrow-derived dendritic cell maturation, the cross-presentation of CD8^+^ T cells and the generation of IFN-γ. Therefore, the immunocompetent BALB/c mice injected with necroptotic DD_RIPK3 cells showed effective anti-tumor responses [58].

Nevertheless, it is also a double-edged sword in cancer biology. On the one hand, it inhibits tumor progression, metastasis and angiogenesis through cancer immunosurveillance [59]. Damage-associated molecular patterns (DAMPs)—molecules released from necroptotic cells—initiate the maturation of dendritic cells and phagocytosis; they also present tumor-specific peptides to CD8^+^ T lymphocytes, bridging innate and adaptive immunity [60]. RIP3 has been reported to activate natural killer T (NKT) cells through phosphatase phosphoglycerate mutase 5 (PGAM5) in a necroptosis-independent manner. A previous experiment indicated that TLR3—a major receptor that also triggers necroptosis—directly induces IL-1 α production in cervical cancer cells, leading to necroptosis [61].

Another study illustrated that dying cancer cells induce DC-mediated IL-12 secretion, which contributes to positive feedback on necroptosis and the elimination of cancer cells [62]. Lalaou et al. indicated that necroptotic cells present tumor-specific antigens (TSAs) or tumor-associated antigens (TAAs) to dendritic cells (DCs) and also release cytokines to trigger the immune response. DCs can also contribute to the maturation of CD8^+^ T lymphocytes to execute anti-tumor responses [63]. Werthmöller et al. reported that B16-F10 melanoma cells treated with the pan-caspase inhibitor z-VAD-fmk and other treatments, such as radiotherapy (RT) and CT, are suppressed owing to the induction of necroptosis and release of high mobility group box 1 protein (HMGB1), which further facilitates the migration of maturing DCs. Furthermore, necroptosis also reduces the infiltration of regulatory T cells (Tregs) and triggers the activation of CD8^+^ T lymphocytes [64].

In addition, emerging evidence has shown that the dysregulation or downregulation of necroptosis-associated proteins is highly correlated with tumor progression [59]. For example, RIP3 and CYLD are downregulated in chronic lymphocytic leukemia (CLL) because lymphoid enhancer-binding factor 1 (LEF1), which is a transcriptional repressor of CYLD, is upregulated [65]. RIP3 is markedly decreased, resulting in NF-κB activation, which promotes inflammation and the survival of cancer cells in patients with acute myeloid leukemia (AML) [66]. RIP3 is significantly decreased in esophageal squamous cell carcinoma, which is chemoresistant to cisplatin [67]. Additionally, under hypoxic conditions but not when it is epigenetically modified by hypermethylation, RIP1 is evidently decreased in the colon carcinoma cell lines HT29, SW480, HCT116, Colo205 and HCT116 [68]. He et al. suggested that the suppression of MLKL expression is related to poor prognosis in patients with ovarian cancer [69]. In osteosarcoma (OS), RIP1 expression is suppressed due to the elevation of miR-155-5p [70]. Koo et al. have shown that genomic methylation adjacent to the transcriptional starting site leads to a decrease in RIP3 in breast cancer patients; moreover, this causes chemoresistance to cisplatin, camptothecin (CPT) and 5-fluorouracil (5-FU) [71].

On the other hand, it also leads to the release of inflammatory sources and an increase in ROS; both of these effects cause genetic instability, favoring cancer development and progression [72]. Based on the mechanism of necroptosis mentioned above, tumor progression is promoted when necroptosis-related proteins are aberrant. Nevertheless, there are potential therapeutic strategies that can be used to evaluate cancer cells and induce necroptotic cell death; these strategies will be discussed in the following section.

### 4.3. The Promotion of Cancer Progression by Necroptosis

Emerging evidence has reported that necroptosis, in addition to suppressing cancer, can also promote cancer progression. CD95 (also called Fas), a necroptotic signaling pathway receptor, has been reported to promote carcinogenesis in mice; moreover, CD95L (also known as FasL) is frequently elevated in cancer patients. However, the loss of CD95 is a good prognostic indicator in liver cancer and ovarian cancer [73].

Liu et al. showed that the ability of necroptotic genes that upregulate cytokine production to promote cancer development outweighs their ability to promote necroptotic cell death. Furthermore, elevated MLKL phosphorylation is positively correlated with poor prognosis in patients with colon and esophageal cancer [74]. Another study indicated that RIP1 overexpression facilitates the proliferation of melanoma cells and leads to anchorage-independent cell growth via NF-κB activation; conversely, melanoma cell proliferation is inhibited by RIP1 silencing [75]. Mechanistically, RIP1 also plays a role in the initiation of the NF-κB signaling pathway; moreover, accumulating studies have indicated that cytokine release and inflammation trigger the proliferation, survival and metastasis of cancer cells. NF-κB disturbs the synthesis and activation of p53, favoring tumor promotion [76]. NF-κB activation and its downstream cytokines also link cancer cells to tumor-associated macrophages (TAMs), which prevent cancer cell attacks from NKT cells and immunosurveillance [72]. In addition, the primary initiator of necroptosis—TNF-α—can stabilize Snail, a critical transcription factor that regulates epithelial-to-mesenchymal transition (EMT), and promote cell migration and metastasis [72]. Seifert et al. suggested that major components of the necrosome, RIP1 and RIP3, are overexpressed and suppress macrophage-induced adaptive immunity through CXCL1 and Mincle signaling, an innate immune response, in pancreatic ductal adenocarcinoma (PDA) [77].

It is evident that the downregulation of RIP1, RIP3 and MLKL is associated with the inhibition of cancer cell growth and increased sensitivity to RT in breast cancer; in addition, necrosulfonamide (NSA), an inhibitor of necroptosis, leads to significant tumor growth suppression [74].

Although DAMPs released from necroptotic cells can induce inflammatory cytokines and antigen presentation to activate CD8^+^ T lymphocytes and thus eliminate cancer cells, they also promote inflammation, angiogenesis, proliferation and metastasis. In addition, inflammation-triggered ROS combined with necroptosis-mediated ROS damage DNA and cause genomic instability, facilitating tumorigenesis [78]. For instance, RIP3 can directly induce TNF-mediated ROS production in NIH 3T3 mouse fibroblast cells.

In the pancreases of RIPK3-knockout p48^Cre^;Kras^G12D^ mice, the expression of programmed death-ligand 1 (PD-L1), a critical ligand that negatively modulates T lymphocyte activation, on macrophages is significantly reduced [78]. Strilic et al. showed that human and murine tumor cells promote extravasation—a critical step in metastasis—in endothelial cells, primarily through necroptosis. Additionally, in mice treated with a RIP1 inhibitor (e.g., Nec-1) or in which RIP3 is silenced, tumor cell extravasation and metastasis are suppressed [79].

## 5. Crosstalk between ROS and Necroptosis

Previous studies have suggested that there is a correlation between necroptosis and ROS under pathophysiological conditions (Figure 3). For instance, curcumol, which is derived from the roots of *Rhizoma curcumae*, can trigger the JNK signaling pathway and increase ROS generation in hepatic stellate cells via RIP1/RIP3-dependent necroptosis [80]. Another study conducted by Zhang showed that the ROS-mediated modification of the cysteine (C) 257, C268 and C586 residues of RIP1 can result in the formation of an intramolecular disulfide bond, which facilities RIP1 autophosphorylation at serine (S) 161 and, in turn, recruits RIP3 to form the necrosome. This indicates that TNF-induced mitochondrial ROS serve as an initiator of necroptosis [81].

Hence, the evidence presented above suggests that a relationship exists between ROS and necroptosis and that they are positively correlated. Lu et al. indicated that RIP1 or RIP3 blockade induced by Nec-1 and GSK872, respectively, not only reduces shikonin-mediated necroptosis in glioma cells but also decreases the generation of intracellular ROS. MnTBAP, a superoxide dismutase mimetic, inhibits ROS production and impedes RIP1 and RIP3 expression [86,87]. In addition, the removal of ROS by butylated hydroxyanisole (BHA), an antioxidant, significantly abrogates TNF-mediated necroptosis in the mouse fibrosarcoma L929 cell line [88].

Moreover, crosstalk between necroptosis and ROS occurs through metabolic signaling. RIP3 serves as an important modulator of several primarily metabolic enzymes: (1) Glycogen phosphorylase (PYGL) activates pyruvate dehydrogenase (PDH), which induces aerobic respiration and oxidative respiration, resulting in an increase in ROS production. Additionally, RIP3-mediated ROS generation facilitates necrosome formation and has a positive effect on necroptosis [84]. (2) Glutamate-ammonia ligase (GLUL) and glutamate dehydrogenase 1 (GLUD1) upregulate glutaminolysis, through which glutamine is converted to glutamate, increasing ROS generation [78,85].

Like RIP3, RIP1 is also associated with ROS generation. Mitochondrial adenine-nucleotide translocase (ANT), an enzyme which is located at the inner mitochondrial membrane, has been reported to play a key role in the modulation of ROS production by regulating the levels of ADP and ATP. RIP1 downregulates the activity of ANT, which in turn leads to the accumulation of ATP and a decrease in ATP synthase activity; this latter effect augments ROS generation [83,89]. Hence, RIP1 activation during necroptosis attenuates ANT to increase ROS levels.

However, in addition to mitochondrial ROS production, the extramitochondrial ROS production, such as that by NADPH oxidase 1 (NOX1), which promotes ROS generation through TNFα stimulation, has been shown to trigger necroptosis [82].

## 6. Potential Therapies Targeting ROS Modulation and Necroptosis

As discussed above, the effectiveness of several anti-tumor drugs is limited by resistance to apoptosis; thus, the modulation of necroptosis and ROS seems to be a potential means for cancer treatment. Previous studies have reported that necroptosis serves as a double-edged sword in cancer progression and development. On the one hand, the induction of necroptosis bypasses apoptosis and triggers cell death through an alternative mechanism in apoptosis-resistant cancer cells. Moreover, preliminary evidence has shown that necroptotic cell death can be elicited in most common cancers (e.g., colorectal cancer, lung cancer, ovarian cancer, breast cancer, hepatocarcinoma, bladder carcinoma and cervical cancer) by necroptosis inducers; thus, various therapies targeting necroptosis have been used in clinical applications and research [2,68]. On the other hand, it evokes inflammation, facilitating the proliferation and metastasis of remaining cancer cells [59,90]. Emerging evidence indicates that ROS are highly correlated with necroptosis; moreover, the modulation of ROS is considered an anti-cancer therapy [91,92]. Thus, killing cancer cells via necroptosis and ROS seems to be a potential strategy.

The following are examples of promising treatments targeting necroptosis and ROS. They are summarized in the tables below (Table 1. and Table 2.) based on their associated molecular mechanisms.

## 7. Treatments Targeting Necroptosis

Here, we summarize a panel of anti-cancer compounds that may potentially target necroptosis (Figure 4). According to Kong et al., at low concentrations, 3u—a novel naphthyridine derivative—upregulates death receptors (DRs) and adaptor proteins (e.g., TRADD) in A375 melanoma cells, triggering necroptosis; nevertheless, at high concentrations, 3u induces caspase-3-dependent apoptosis through caspase-8 activation and the upregulation of DRs and scaffold protein [93].

The LNCaP-AI cell line, a castration-resistant prostate cancer cell line, is evidently resistant to hormone therapies, and Polo-like kinase 1 (Plk1) expression is increased in LNCaP-AI cells compared with in LNCaP cells. Plk1 plays a role in mitosis progression [123,124]. Based on an experiment by Deeraksa, BI2536—a small selective molecule inhibitor of Plk1—inhibits cell growth via necroptosis instead of apoptosis, and this was confirmed by the use of Nec-1 [94]. Moreover, another study also suggested that BI2536 activates poly (ADP-ribose) polymerase-1 (PARP-1), resulting in DNA damage and fragmentation rather than caspase-dependent cell death [94,125].

Ceramide, an important intermediate in sphingolipid metabolism in mammals, plays a role in intracellular signaling pathways, such as modulating cell differentiation, proliferation and apoptosis [126,127]. Ceramide is also being investigated in Phase I clinical trials in cancer treatment [95]. Zhang et al. illustrated that ceramide nanoliposomes (CNLs), which consist of a nanoscale delivery system and ceramide, can significantly cause cell death in cisplatin-resistant ovarian cell lines (e.g., A2780cp and PE04) through necroptosis and the induction of MLKL [95].

Hypoxia is a microenvironment that facilitates angiogenesis and anti-cancer events. Cobalt chloride is reported to stabilize HIF1α in cell culture and is used to generate a hypoxic environment [128,129,130]. Wang et al. showed that in the human colon cancer cell line HT-29, cobalt chloride can induce a significant increase in RIPK1, RIPK3 and MLKL in the presence of aberrant caspases, initiating necroptosis; thus, it may serve as a potential treatment for cancer cells that are resistant to apoptosis [96]. Fingolimod (FYT720), a new FDA-approved drug for multiple sclerosis, directly binds sI2PP2A/SET, converting inactive protein phosphatase 2A (PP2A), a tumor suppressor enzyme, to the active form; this upregulates RIP1 and induces necroptosis to suppress A549 lung tumor cells both in vitro and also in vivo SCID mice implanted by xenografts of A549/sh-I2PP2A/SET [97,131].

2-methoxy-6-acetyl-7-methyljuglone (MAM) is a natural naphthoquinone isolated from the Chinese medicinal plant *Polygonum cuspidatum*; it has been reported to deplete glutathione (GSH)/GSH disulfide (GSSG) levels, inhibit ROS generation and activate the JNK signaling pathway. It subsequently induces NO-dependent necroptosis in A549 human lung cancer cells [98]. Another study showed that MAM induces necroptosis in cisplatin-resistant A549 cells and H1299 cells by targeting RIP1 instead of the TNFα signaling pathway; furthermore, it can also lead to the suppression of the growth of A549 cells in a xenograft nude mouse model upon the intraperitoneal injection of the MAM [99]. Sun et al. suggested that MAM initiates RIP1 and RIP3 phosphorylation and subsequently activates JNK to augment he mitochondrial ROS production mediated by cytosolic calcium in HCT116 and HT29 colon cancer cells [100].

Metformin is a standard orally administered drug used to reduce blood sugar levels in diabetes mellitus patients [132], and simvastatin serves as an HMG-CoA reductase inhibitor for patients with hypercholesterolemia [133]. However, Babcook suggested that cotreatment with metformin and simvastatin can trigger the necroptosis of C4-2B metastatic castration-resistant prostate cancer cells by upregulating RIP1 and RIP3 and inhibit cell cycle arrest at the G1 phase [101].

A study by F. Basit showed that Obatoclax (GX15-070), a small-molecule antagonist of anti-apoptotic Bcl-2 proteins, leads to necroptosis in rhabdomyosarcoma (RMS) in a chicken chorioallantoic membrane (CAM) model by inducing the assembly of the necrosome, which consists of RIP1, RIP3 and MLKL, on the autophagosomal membrane [102].

Shikonin (SK), a naphthoquinone compound extracted from *Lithospermum erythrorhizon* and the enantiomer of alkannin, triggers necroptosis by upregulating RIP1 and RIP3 [87,134]. Chen et al. reported that SK leads to AsPC-1 human pancreatic tumor cell death through necroptosis; additionally, SK can improve the efficacy of gemcitabine in specific pathogen-free female nude mice (athymic, Balb/c nu/nu) [54]. Han et al. indicated that SK-induced necroptosis causes the death of Bcl-2- or Bcl-xL-overexpressing MCF-7 breast cancer cells, probably by downregulating procaspase-8; nevertheless, this process requires further investigation. Moreover, an in vivo experiment was also conducted: nude female mice that were injected with MCF-7 cancer cells showed decreased weights of tumors under SK administration [104]. An experiment by Wada suggested that SK can cause necroptotic cell death in various multiple myeloma (MM) cell lines (e.g., KMS-12-PE, RPMI-8226 and U266) through the downregulation of caspase-3/8 and upregulation of RIP1 [105].

Hammerová et al. reported that sanguilutine (SL), a potent inducer of caspase-independent non-apoptotic death, primarily induces melanoma cell death via necroptosis and synergistically decreases cancer cell viability with an autophagy inhibitor [106]. The human melanoma A375 cell line and Mel-JuSo cell line simultaneously treated with SL and Nec-1 showed significant reductions in the percentages of dead A375 and Mel-JuSo cells. This indicated that necroptosis was the predominant form of cell death in melanoma cells triggered by SL, leading to speculation that necroptosis was induced through RIPK1 [106].

Sorafenib is an orally administered multikinase inhibitor of vascular endothelial growth factor receptor (VEGFR) that is approved for the treatment of advanced hepatocellular and renal carcinoma [65,135,136]. The necroptosis of Z-VAD-fmk-plus-Nec-1-treated MM.1S cells, a human multiple myeloma cell line, can be induced when caspases are inefficiently activated [107]. Therefore, it is suggested that Sorafenib can not only trigger Puma-mediated apoptosis but also induce necroptosis [107].

Staurosporine (STS), an antifungal agent that markedly inhibits phospholipid/Ca^2+^-dependent protein kinase C (PKC), is derived from *Streptomyces staurosporeus* [137,138]. It has been reported to induce apoptosis in human breast cancer MCF-7c3 cells [139]. Moreover, it has also been indicated to trigger necroptosis via the upregulation of RIP1 and MLKL in human lymphoma U937 cells in the presence of aberrant caspases; however, necroptosis can be partially circumvented by Nec-1 [108].

Tanshinone IIA (Tan IIA), a lipophilic compound of *Salvia miltiorrhiza*, has previously been approved for the treatment of cardiovascular diseases, such as blood clotting abnormalities, atherosclerosis and myocardial infarction [140]. Chang et al. also indicated that it exerts synergistic effects with transresveratrol to increase the cytotoxicity and cell death of human hepatocellular carcinoma HepG2 cells [141]. Additionally, Lin et al. reported that Tan IIA can simultaneously induce both apoptosis and necroptosis in HepG2 cells. cFLIP is evidently downregulated in Tan IIA-treated cells, resulting in the activation of caspase-8, RIP1, RIP3 and MLKL and, subsequently, apoptosis and necroptosis [109]. However, the administration of a pan-caspase inhibitor (e.g., Z-VAD-fmk) results in a switch from apoptosis to necroptosis [142]. Nec-1 can restore the cFLIP levels downregulated by Tan IIA, blocking both apoptosis and necroptosis [109].

Preliminary evidence has indicated that zinc oxide nanoparticles (ZnO NPs) induce the selective cell death of proliferating cells but not normal cells [143]. In addition, Farasat et al. suggested that ZnO NPs can induce necroptosis in MCF-7 breast cancer cells by upregulating RIP1, RIP3 and MLKL, and this was confirmed by the application of z-VAD-fmk and an autophagy inhibitor, 3-MA [110].

## 8. Treatments Targeting Both ROS and Necroptosis

BAY 87-2243 (BAY), which inhibits mitochondrial complex I and increases ROS levels, triggers mitochondrial permeability transition pore (mPTP) opening, autophagosome formation and mitophagy; this elevates ROS generation, resulting in RIP1/MLKL upregulation and the necroptotic cell death of BRAFV600E melanoma cells [111].

Bufalin is a major component of Chan Su, a traditional Chinese medicine, and is used to promote apoptosis in some human leukemia cells and MGC803 gastric cancer cells [144,145]. Furthermore, the intravenous injection of Bufalin has been approved as an adjuvant for patients with hepatocellular carcinoma (HCC) and non-small cell lung cancer (NSCLC) in China [146]. However, Bufalin-induced necroptosis is less investigated. Li et al. reported that Bufalin can induce ROS accumulation and RIP1-dependent necroptosis by upregulating the RIP1/RIP3/PGAM5 pathway in the breast cancer cell lines MDA-MB-468 and T47D [112].

Dimethyl fumarate (DMF), an FDA-approved immunomodulatory therapy for multiple sclerosis (MS), has been suggested to induce necroptosis in colon cancer CT26 cells [113,114]. DMF induces cellular GSH depletion; this causes ROS generation, subsequent MAPK signaling pathway activation and mitochondrial depolarization, resulting in necroptotic cell death [114].

Docetaxel, a taxane anti-cancer drug, is considered a first-line treatment for triple-negative breast cancer (TNBC) and has also been approved for the treatment of breast cancer and non-small-cell lung cancer [147,148]. However, previous studies have shown that the efficacy of taxane is limited by resistance (e.g., spindle assembly checkpoint (SAC) defects) [149]. BAD, a Bcl-2-associated death promoter, serves as a prognostic indicator of the effectiveness of docetaxel [150]. ROS-dependent BAD-induced necroptotic death evidently sensitizes MDA-MB-231 TNBC cells to docetaxel [115]. In addition, Mann et al. also suggests that BAD sensitizes breast cancer tumor xenografts to docetaxel in the nude mice [115].

Deoxypodophyllotoxin (DPT) is a natural microtubule destabilizer [151]. It has been reported to impede tubulin polymerization—a pivotal process during mitosis—and to initiate G2/M cell cycle arrest in HeLa cells [152]. In addition, DPT can induce necroptosis by increasing the ROS generation and decreasing the mitochondria membrane potential in NCI-H460 (H460) human non-small-cell lung cancer cells and in vivo xenograft mouse models [116].

A study by Mármol revealed that gold (I) complex (Au(C≡C-2-NC5H4) (PTA) can enter and interfere with the physiological functions of mitochondria, resulting in the accumulation of ROS. Moreover, ROS also activate NF-κB and lead to increased TNF-α generation, forming a positive feedback loop; both effects induce RIP1-dependent necroptotic cell death in Caco-2 colorectal adenocarcinoma cells [117].

C/EBP homologous protein (CHOP) is a proapoptotic transcription factor and a downstream component of the unfolded protein response (UPR) that contributes to cell death [153]. LGH00168, a novel CHOP activator, has been suggested to induce the loss of the mitochondrial membrane potential (Δψm), mitochondrial ROS generation, ER stress, NF-κB inhibition and RIP1-dependent necroptosis in A549 cells and lung carcinoma-bearing mice [118].

Neoalbaconol, a small-molecular compound derived from the fungus *Albatrellus confluens*, has been indicated to inhibit cancer progression in several cancer cell lines [154]. For instance, it decreases the growth and viability of the breast cancer cell lines MDA-MB-231, MCF-7 and MX-1. In addition, it inhibits VEGF-mediated angiogenesis in C57BL/6 mice [155]. An experiment conducted by Yu suggested that neoalbaconol can induce autocrine TNFα and ROS production, which initiates RIP1- and RIP3-dependent necroptosis in C666-1 cells, HK-1 cells (nasopharyngeal carcinoma cell lines), MX-1 cells (a human breast cancer cell line) and AGS-EBV cells (a human gastric cancer cell line) [55].

Cotreatment with poly (I:C), a synthetic dsRNA-mimic, and z-VAD-fmk, a pan-caspase inhibitor, can lead to CT26 mouse colon carcinoma cell death through necroptosis as well as TLR3/TICAM1-mediated ROS production. In addition, cell death is inhibited by treatment with Nec-1 or RIP3 knockdown; this indicates that cotreatment with CT26 and poly (I:C) can induce necroptosis in vitro and in an in vivo mouse model [119].

Resibufogenin, a bufadienolide compound, is a Chinese herbal medicine extracted from toad venom. Han et al. suggested that it can initiate necroptosis to suppress metastasis in both human CRC HCT116 and SW480 cells and in an in vivo xenograft mouse model *through* the elevation of RIP3. In addition, it also promotes the expression of PYGL, GLUD1 and GLUL, which are important enzymes, via RIP3 to generate ROS [120].

With advances in technology, the combination of nanoparticles with anti-cancer drugs has shown increased efficacy and reduced side effects and is therefore seen as a potential strategy for cancer therapy [156]. Sonkusre et al. showed that selenium (Se), an essential trace mineral in the human body, incorporates with nanoparticles to activate the TNF signaling pathway and promote ROS generation, both of which lead to necroptosis, in a human prostate adenocarcinoma cell line (PC-3) [121].

Tanshinol (3-(3′,4′-dihydroxyphenyl)-2-hydroxypropanoic acid), a bioactive monomer from the roots of the traditional Chinese medicinal herb *Salvia miltiorrhiza* (Danshen), can protect endothelial cells from homocysteine-induced injury and is also a treatment for cardiovascular disease (e.g., angina pectoris) [157,158]. Additionally, it has been reported to induce the MLKL-mediated necroptosis of A549 and H1299 cells through RIP1- and RIP3-independent ROS generation [122].

## 9. Conclusions and Perspectives

In this review, we first elucidated the roles of necroptosis and ROS in human physiology. Necroptosis, as a double-edged sword, can modulate immunity and play a role in cancer biology; ROS can trigger apoptosis and the inflammatory response (e.g., T lymphocyte activation). Moreover, we shed light on the correlation between necroptosis and ROS on a molecular basis; they can form a positive feedback loop, allowing mutual activation. On the one hand, extramitochondrial ROS generated by NOX1 and mitochondrial ROS can initiate the activation of RIP1 and the recruitment of RIP3 to induce necroptosis. On the other hand, RIP3 can upregulate aerobic respiration to augment ROS production via metabolic pathways (e.g., PYGL, PDH and GLUD1). Owing to emerging anti-apoptotic chemoresistance, alternative therapies should be more emphasized and further investigated. Here, we illustrated and summarized several promising anti-cancer treatments targeting necroptosis and ROS. Although the molecular mechanism underlying the relationship between necroptosis and ROS has been established, effective therapies to inhibit cancer progression via the simultaneous modulation of necroptosis and ROS require further investigation. In addition, more research should be conducted to identify which cancer cell lines can be considerably inhibited by necroptosis and ROS.

## Figures and Tables

**Figure 1 cancers-12-02185-f001:**
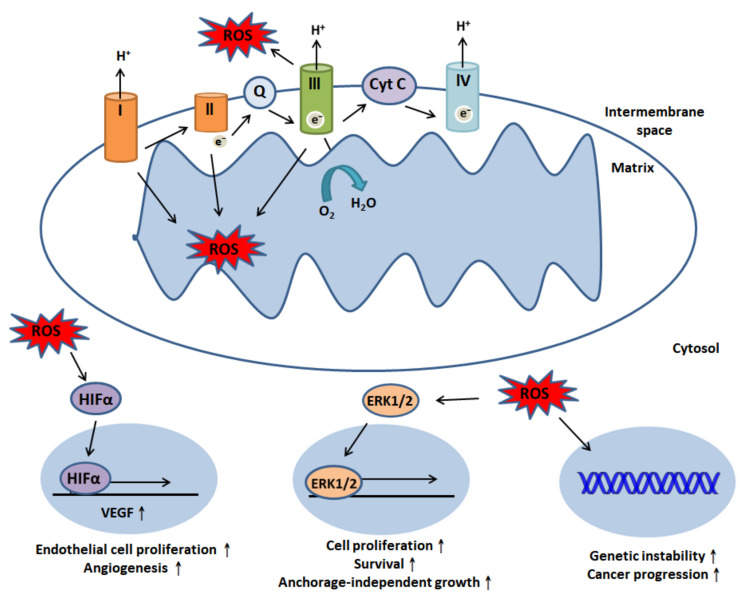
Reactive oxygen species (ROS) generation and modulation of cell signaling events. The four mitochondrial complexes embedded in the mitochondrial inner membrane—complexes I, II, III and IV—are responsible for electron transfer. ROS can be generated by complexes I, II and III. ROS produced by complexes I and II are primarily located in the matrix, while complex III can generate O2• in both the intermembrane space and matrix. Nevertheless, intermembrane ROS translocate to the cytosol and cellular trigger events; ROS can stabilize HIFα under hypoxic conditions, and this stabilization subsequently induces VEGF expression to promote endothelial cell proliferation and angiogenesis. ROS can also initiate the ERK signaling pathway, promoting cell proliferation and survival and anchorage-independent growth. Moreover, they can lead to genomic instability and induce cancer progression.

**Figure 2 cancers-12-02185-f002:**
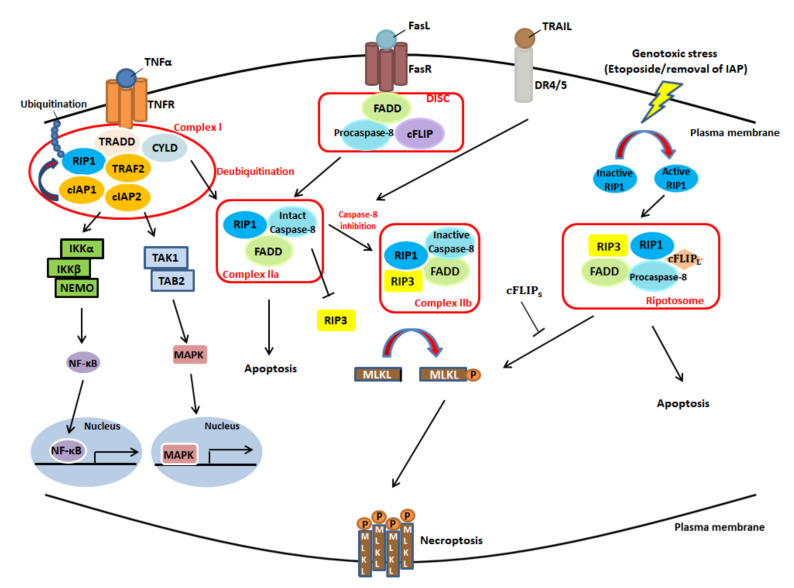
Overview of the cell death receptor-associated signaling pathway. The signaling pathway is initiated by the interaction between the TNFR superfamily members and their ligands. (**a**) TNF-α binds with TNFR1, which recruits TRADD, TRAF2, CYLD, cIAP1, cIAP2 and RIP1 to form complex I. If cIAP1 polyubiquitinates RIP1, complex I activates both the NF-κB and MAPK signaling pathways. However, if RIP1 is deubiquitinated by CYLD, complex IIa or complex IIb may be subsequently formed to induce apoptosis or necroptosis, respectively. When caspase-8 is intact, the formation of complex II, which leads to apoptosis and RIP3 inhibition, is favored. Conversely, if caspase-8 is compromised, the formation of complex IIb (known as the necrosome), which consists of RIP1, RIP3 and FADD, is inclined to phosphorylate MLKL, leading to necroptosis via the translocation of phosphorylated MLKL and plasma membrane permeabilization. (**b**) Fas ligand (CD95L) binds with Fas receptor (CD95); this recruits FADD, procaspase-8 and cFLIP to form DISC. Similarly, intact caspase-8 favors the formation of complex IIa and apoptosis. However, inactive caspase-8 contributes to the formation of complex IIb and necroptosis. (**c**) The interaction between TRAIL and DR4/5 leads to the apoptotic and necroptotic situations mentioned above, based on the activity of caspase-8. (**d**) In the presence of genotoxic stresses, the removal of IAP or anti-tumor drugs (e.g., etoposide), RIP1 is activated to form the ripotosome, which consists of RIP1, RIP3, FADD, procaspase-8 and ${\mathrm{cFLIP}}_{L}$. If ${\mathrm{cFLIP}}_{s}$ is present, the phosphorylation of MLKL by the ripotosome is inhibited, and apoptosis is favored. By contrast, the absence of ${\mathrm{cFLIP}}_{s}$ leads to the phosphorylation of MLKL, which favors necroptosis. The figure was redrawn and modified from [2].

**Figure 3 cancers-12-02185-f003:**
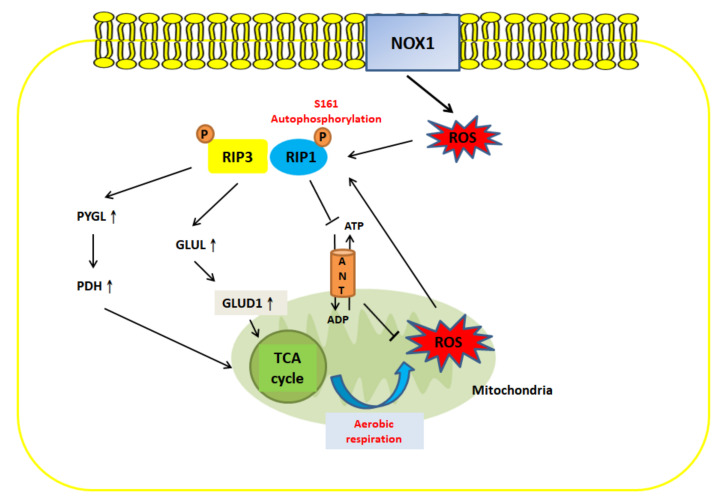
The crosstalk between ROS and necroptosis. ROS and necroptosis can form a positive feedback loop. Extramitochondrial ROS production by NOX1 or mitochondrial ROS generation through oxidative phosphorylation modifies the C257, C268 and C586 residues of RIP1 and facilitates the autophosphorylation of RIP1 at S161. This leads to the recruitment of RIP3 and necroptosis [81,82]. RIP1 can induce ROS production by inhibiting adenine-nucleotide translocase (ANT) and increasing ATP [83]. RIP3 can induce ROS generation through metabolic signaling pathways: (a) Upregulation of PYGL promotes the expression of PDH, which increases TCA cycle activity to induce aerobic respiration and produce ROS [84]. (b) Elevation of GLUL upregulates GLUD1 and increases glutaminolysis, further facilitating the TCA cycle and aerobic respiration. Furthermore, RIP1- and RIP3-mediated ROS production can reactivate necroptosis, forming a positive feedback loop [78,85].

**Figure 4 cancers-12-02185-f004:**
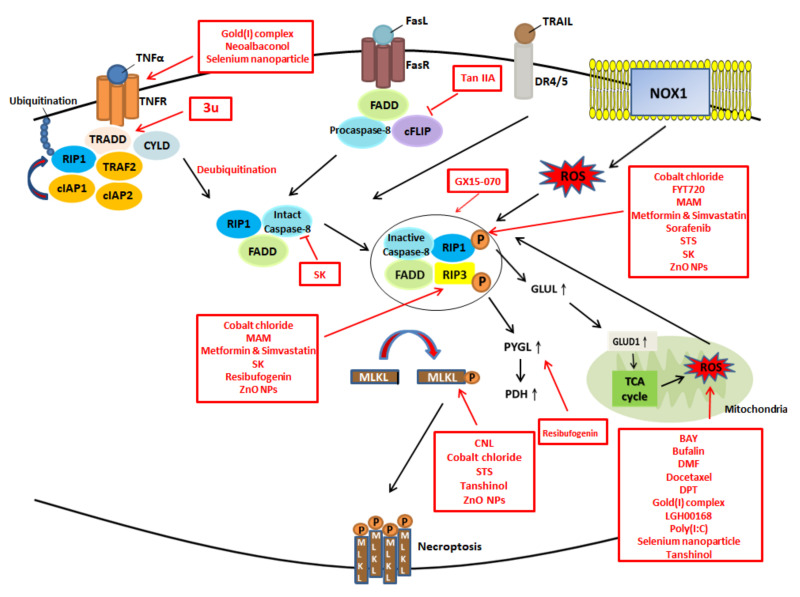
The potential drugs and their targets in modulation of necroptosis and ROS.

**Table 1 cancers-12-02185-t001:** Treatments targeting necroptosis.

Drug	Mechanism	Cell Line/In Vivo	Reference
3u (naphthyridine derivative)	Upregulation of death receptors and adaptor proteins (e.g., TRADD)	Melanoma cells A375	[93]
BI2536 (2-aminopyrimidine-containing compound)	Inhibition of Plk1 and induction of necroptosis	Castration-resistant prostate cancer LNCaP-AI	[94]
Ceramide nanoliposomes (CNL)	Induction of MLKL	Cisplatin-resistant ovarian cancer cell lines A2780cp and PE04	[95]
Cobalt chloride	Significant increase in RIPK1, RIPK3 and MLKL	Colon cancer cell HT29	[96]
FYT720 (Fingolimod)	Upregulation of RIP1	NSCLC A549/A549/sh-l2PP2A/SET xenograft SCID mice	[97]
MAM (natural naphthoquinone)	Induction of NO-dependent necroptosis	NSCLC A549	[98]
	Targeting RIP1 instead of TNFα signaling pathway	NSCLC A549 and H1299/A549-derived xenograft nude mice	[99]
	Increased RIP1 and RIP3 phosphorylation	Colon carcinoma HCT116 and HT29	[100]
Metformin and Simvastatin	Upregulation of RIP1 and RIP3	Metastatic castration-resistant prostate cancer cells C4-2B	[101]
Obatoclax (GX15-070)	Inducing the assembly of necrosome	Rhabdomyosarcoma RMS	[102]
Shikonin	Upregulation of RIP1 and RIP3	Pancreatic tumor cell AsPC-1/in vivo	[103]
	Downregulation of caspase-8	Breast adenocarcinoma cell MCF-7/ nude female mice injected with MCF-7	[104]
	Downregulation of caspase-3/8 and upregulation of RIP1	Multiple-myeloma cell lines KMS-12-PE, RPMI-8226 and U266	[105]
Sanguilutine (SL)		Melanoma cell lines A375 and Mel-JuSo	[106]
Sorafenib	Upregulation of RIP1	Multiple-myeloma cell line MM.1S	[107]
Staurosporine (STS)	Upregulation of RIP1 and MLKL	Lymphoma cell line U937	[108]
Tanshinone IIA (Tan IIA)	Downregulation of cFLIP	Hepatocellular carcinoma cell HepG2	[109]
ZnO NPs	Upregulation of RIP1, RIP3 and MLKL	Breast adenocarcinoma cell line MCF-7	[110]

**Table 2 cancers-12-02185-t002:** Treatments targeting both ROS and necroptosis.

Drug	Mechanism	Cell Line/In Vivo	Reference
BAY 87-2243 (BAY)	Elevated ROS generation and upregulation of RIP1/MLKL	BRAFV600E melanoma cell lines	[111]
Bufalin	Induction of ROS and RIP1-dependent necroptosis by upregulating RIP1/RIP3/PGAM5 pathway	Breast cancer cell lines MDA-MB-468 and T47D	[112]
Dimethyl fumarate (DMF)	Glutathione (GSH) depletion, increased ROS generation and induction of necroptosis	Murine colon cancer cell CT26	[113,114]
Docetaxel	BAD-induced necroptotic death depending on ROS	Breast cancer cells MDA-MB-231/in vivo	[115]
Deoxypodophyllotoxin (DPT)	Increased ROS generation	NSCLC NCI-H460/in vivo	[116]
Gold(I) complex	Accumulation of ROS and generation of TNF-α, leading to RIP1 activation	Colorectal adenocarcinoma cells Caco-2	[117]
LGH00168	Mitochondrial ROS generation, ER stress, NF-κB inhibition and RIP1-dependent necroptosis	NSCLC A549/in vivo	[118]
Neoalbaconol	Inducing autocrine TNFα and ROS production, and initiating RIP1 and RIP3-dependent necroptosis	Nasopharyngeal carcinoma cell lines C666-1 and HK-1, breast cancer cell line MX-1, human gastric cancer cell AGS-EBV	[55]
Poly(I:C)	Induction of necroptosis as well as ROS production through the production of TLR3/TICAM1-mediated ROS	Murine colon cancer cell CT26/in vivo	[119]
Resibufogenin	Elevation of RIP3, PYGL, GLUD1 and GLUL to induce both necroptosis and ROS generation	Colon carcinoma cell line HCT116 and SW480/in vivo	[120]
Selenium nanoparticle	Activation of TNF signaling pathway and promotion of ROS generation	Prostate adenocarcinoma cells PC-3	[121]
Tanshinol	Induction of MLKL-mediated necroptosis and ROS generation	NSCLC H1299 and A549	[122]

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
