# Peer review of "The Role of Necroptosis in ROS-Mediated Cancer Therapies and Its Promising Applications"

_cancers, 2020, doi:10.3390/cancers12082185_

Round 1
Reviewer 1 Report
We thank the authors for answering this reviewer’s questions in full– in text and Table 1 and by adding the new figure.
Some minor layout issues remain (should be easily resolved in the final processing).
The only remaining point is, that (the new) sections 4.2 and 4.3 are still overlapping – albeit having different perspectives on the same molecules and cellular mechanisms.
Reviewer 2 Report
Hsu et al. present a thoroughly revised manuscript which addresses all my comments. The work reads well, is clear and balanced, and provides new insights into the role of necroptosis in ROS-mediated cancer
Reviewer 3 Report
I have checked the revision and the response. The authors have response all to my comments. I think the revised version is suitable to publish in your journal.
This manuscript is a resubmission of an earlier submission. The following is a list of the peer review reports and author responses from that submission.
Round 1
Reviewer 1 Report
This manuscript gives a nice review of necroptosis, linking it to ROS. Then a nice connection onto how various substances and drugs are interacting with different key players in these pathways is made.
Examples of many drugs acting on various cell lines are given
Are there in addition to the in vitro experiments in vivo studies?
For only one drug Sorafenib –approval for clinical use in Cancer is mentioned. Is there clinical evidence for other drugs already in use (for other purposes) e. g. metformin , DMF, DPT, herbal extract used in chinese medicine?
Many examples and referenced papers refer to ‚herbal‘ extracts, not standard chemotherapies or mechanistial papers.
A figure connecting the drugs and their targets in relatinship to necrpotosis and ROS would be helpfull.
Reference 2 includes a discussion of apoptosis, but mainly focuses on necroptosis.
In the field of HIF, VEGF, angiogenesis reference 22 is a recent, but not the most obvious choice, a more groundbraking paper should be included.
Section 4.2 is very short and seems to serve as a ‚linker‘, neverthelsess, here it would be helpfull to go somewhat more into depth, or at least refer to recent refernces/reviews.
The titles and partition of section 4.3.and 4.4 are somewhat misleading, since several aspects of the important topic of immunosurveillance are described in both sections. The content of both sections does not focus on its respective titels , but includes many interacting mechanisms. It would be helpfull to segregate those a little bit more clearly. Especially a separate section on metabolism (already included in seciton 4.3.) would improve the manuscript.
With some improvement this manuscript could give a comprehensive overview of necorptosis/ROS targeting substances in cancer.
Author Response
Q1. Examples of many drugs acting on various cell lines are given
Are there in addition to the in vitro experiments in vivo studies?
Ans: We thank the reviewer for the helpful suggestions. The additional information has been added to Table 1 in the revision of the manuscript.
Q2. For only one drug Sorafenib –approval for clinical use in Cancer is mentioned. Is there clinical evidence for other drugs already in use (for other purposes) e. g. metformin, DMF, DPT, herbal extract used in Chinese medicine?
Ans: We thank the reviewer’s suggestion. The relevant information of these drugs has been added to the revision of the manuscript. For instance, Docetaxel has been approved for the treatment of breast cancer and non–small cell lung cancer. Most of the examples have not been approved yet and some have been approved for other disease instead of cancer (e.g., Metformin, FYT720 and DMF). Nevertheless, the examples mentioned in the manuscript are potential for cancer therapies. We also added the relevant descriptions to the section “Treatments Targeting Necroptosis” of the revision of the manuscript.
Q3. Many examples and referenced papers refer to herbal‘extracts, not standard chemotherapies or mechanistically papers.
Ans: We thank the suggestions of the reviewer. In this review, we provided many examples are natural compounds but rather than the extracts with mechanism. Besides, some of the examples are standard therapeutic agents, for example, the Sorafenib and Metformin. We added the detailed to the revision of the manuscript. The examples are as follows,
Q4. A figure connecting the drugs and their targets in relationship to necroptosis and ROS would be helpful.
Ans: We thank the reviewer for the helpful suggestion. We added a new figure (Figure 4 in the revision) connecting the drugs and their targeting in relationship to necroptosis and ROS to the revision of the manuscript accordingly.
Q5. Reference 2 includes a discussion of apoptosis, but mainly focuses on necroptosis.
Ans: We thank for the reviewer’s suggestion. Although the title of reference 2 is associated with necroptosis, the section Proapoptotic therapy (e.g., using cisplatin, carboplatin, paclitaxel, 5-fluorouracil, and gemcitabine), a major form of chemotherapy, is the principal method for cancer treatment’’ to support this sentence.
Q6. In the field of HIF, VEGF, angiogenesis reference 22 is a recent, but not the most obvious choice, a more groundbreaking paper should be included.
Ans: We thank the reviewer’s suggestion and we replaced the reference with two papers [32] and [33] in the revision of the manuscript.
Q7. Section 4.2 is very short and seems to serve as a linker‘, nevertheless, here it would be helpful to go somewhat more into depth, or at least refer to recent references/reviews.
Ans: We thank the reviewer’s helpful suggestions. We merged Section 4.3 to Section 4.2 in the revision of the manuscript.
Q8. The titles and partition of section 4.3.and 4.4 are somewhat misleading, since several aspects of the important topic of immunosurveillance are described in both sections. The content of both sections does not focus on its respective titles, but includes many interacting mechanisms. It would be helpful to segregate those a little bit more clearly. Especially a separate section on metabolism would improve the manuscript.
Ans: We thank the reviewer’s helpful suggestions. To avoid the misleading, we merged Section 4.3 to Section 4.2 in the revision of the manuscript.

Reviewer 2 Report
Hsu et al. present a manuscript reviewing necroptosis and their impact on ROS-mediated cancer therapy. The topic of the review is very interesting, suited for the readership of this journal, and the drug examples cover a variety of broadly relevant areas. The structure of the review is relatively clear. However, in its current form, I have several criticisms that prevent me from supporting acceptance at this stage.
Major criticisms
- Review content – In its current form, the drugs in section 5. Potential therapies targeting ROS modulation and necroptosis is a just collection of examples but does not provide an overarching synthesis of findings. The only new concept is crosstalk between necroptosis and ROS – other broader questions that a synthesis of the different examples include could be: what are the relative contributions of different aspects of drugs (necroptosis controversial effects on cancer) to ROS-mediated cancer regulation? Which pathway appears to be more frequently used? Any general patterns of the drug are used in therapies? What role does necroptosis regulating drug play in inflammation?
- Completeness of Section 4. Necroptosis – It would be helpful if section 4 can discuss two aspects of necroptosis in cancer (promoting vs inhibiting). The paragraph in 4.2 The role of necroptosis in cancer is too short and has no informative description.
- Figures – Section 4. Necroptosis can be summarized in a figure
- References – The review cites many other reviews, throughout the text, even though the text makes it sound as if primary data was described (For example, Reference 11). I would encourage the authors not to cite reviews but focus on the primary papers in these instances.
Author Response
Q1. Review content – In its current form, the drugs in section 5. Potential therapies targeting ROS modulation and necroptosis is a just collection of examples but does not provide an overarching synthesis of findings. The only new concept is crosstalk between necroptosis and ROS – other broader questions that a synthesis of the different examples include could be: what are the relative contributions of different aspects of drugs (necroptosis controversial effects on cancer) to ROS-mediated cancer regulation?
Ans: We thank the comments of the reviewers. Necroptosis has controversial effects on cancers, which may depend on the context of cancer cells or the stage of cancer progression. For example, autophagy flux is an essential for degrading damaged organelles or misfolded proteins to maintain the cellular physiology, and constitutively activation of autophagy flux has been shown to be correlated with the chemoresistance of many cancer cells. However, the disturbance of autophagy flux by anti-cancer drugs is considered as a strategy of sensitizing cancer cells. Likewise, the proper modulation of ROS and necroptosis based on the type or stage of cancer cells could be a promising way to sensitize cancer cells to anti-cancer drugs or could elicit an immunity response to eliminate cancer cells.
Q2. Which pathway appears to be more frequently used? Any general patterns of the drug are used in therapies?
Ans: We thank the comments of the reviewers. Although the present studies showed that several pathways including TGF-a, Wnt/b-catenin and NF-κB which are involved in the induction of necroptosis, which seems favors inflammation-associated signaling pathways. However, still non-inflammation pathways may also modulate necroptosis. We also added some discussions to the revision of the manuscript.
Q3. What role does necroptosis regulate drug play in inflammation?
Ans: We thank the reviewer’s suggestions. We described some roles of necroptosis in inflammation in the revision of the manuscript.
Q4. Completeness of Section 4. Necroptosis – It would be helpful if section 4 can discuss two aspects of necroptosis in cancer (promoting vs inhibiting). The paragraph in 4.2. The role of necroptosis in cancer is too short and has no informative description.
Ans: We are thankful for the reviewer’s helpful suggestions. The paragraph in Section 4.2 is short and correlated with Section 4.3, we integrated Section 4.3 with 4.2 in the revision of the manuscript accordingly.
Q5. Figures – Section 4. Necroptosis can be summarized in a figure
Ans: We thank the reviewer for the helpful suggestion. We added a figure (Figure 4 in the revision) connecting the drugs and their targeting in relationship to necroptosis and ROS to the revision of the manuscript accordingly.
Q6. References – The review cites many other reviews, throughout the text, even though the text makes it sound as if primary data was described (For example, Reference 11). I would encourage the authors not to cite reviews but focus on the primary papers in these instances.
Ans: We thank for helpful comments of the reviewer, we cited more references from the primary papers to the revision of the manuscript.

Reviewer 3 Report
The article reviewed the correlation of ROS, necroptosis, Cell Death Signaling Pathways, Human Physiology and Cancer Biology. And then it introduced the potential therapies targeting necroptosis and ROS. The article is interesting in the correlation of Necroptosis in ROS-mediated cancer therapies. It is good written and informative for readers. I suggest the authors to improve the following points before publication.
- This article could be looked as a review for reviews, since it is summarized mainly from review articles rather than original articles. So, the authors should clarify the advantage points to compare with the related reviews published.
- There is no citation for Figure 3 in the text;
- In Figure 2. Overview of the cell death receptor-associated signaling pathway, it is showed very similar pathway to the Figure 1. Necroptotic pathway in Cancer therapy in the necroptosis era. Cell Death Differ. 2016 May; 23(5): 748–756. (PMID: 26915291). If Figure 2 is not original, it is better to insert the cited source or revised source.
Author Response
Q1. This article could be looked as a review for reviews, since it is summarized mainly from review articles rather than original articles. So, the authors should clarify the advantage points to compare with the related reviews published.
Ans: We thank the comments and the suggestions of the reviewers. We updated the cited references which are original articles rather than review articles in the revision of the manuscript.
Q2. There is no citation for Figure 3 in the text;
Ans: We thank the reviewer for pointing out the neglection. The citation for Figure 3 has been added to the text of the revision of the manuscript.
Q3. In Figure 2. Overview of the cell death receptor-associated signaling pathway, it is showed very similar pathway to the Figure 1. Necroptotic pathway in Cancer therapy in the necroptosis era. Cell Death Differ. 2016 May; 23(5): 748–756. (PMID: 26915291). If Figure 2 is not original, it is better to insert the cited source or revised source.
Ans: We thank the helpful suggestions and comments of the reviewer. The cited source (Cell Death Differ. 2016 May; 23(5): 748–756) has been inserted to the legend of Figure 2 in the revision of the manuscript.
